# Lost in Translation: Benchmarking Commercial Machine Translation Models for Dyslexic-Style Text

**Gregory Price**
School of Electrical Engineering and Computer Science
University of Ottawa
gpric024@uottawa.ca

**Shaomei Wu**
AImpower.org
Mountain View, CA 94043
shaomei@aimpower.org

## Abstract

Dyslexia is a neurodivergence that impacts one's ability to process and produce textual information. While previous research has identified unique patterns in the writings of people with dyslexia - such as letter swapping and homophone confusion - that differ themselves from the text typically used in the training and evaluation of common natural language processing (NLP) systems such as machine translation (MT), it is unclear how current state-of-the-art NLP systems perform for users with dyslexia. In this work, we explore this topic through a systematic audit of the performance of commercial MT services using synthetic dyslexia data. By injecting common dyslexia-style writing errors into popular benchmarking datasets, we benchmark the performance of three commercial MT services and one large language model (LLM) with various types and quantities of dyslexia-style errors and show a substantial disparity in MT quality for dyslexic and non-dyslexic text. While people with dyslexia often rely on modern NLP tools as assistive technologies, our results shed light on the fairness challenges experienced by this demographic with popular NLP services, highlighting the need to develop more inclusive and equitable NLP models for users with diverse language use patterns.

## 1  Introduction

Dyslexia is one of the most common learning disabilities, estimated to affect 10% to 17% of English speaking population [33, 3]. As a neuro-cognitive condition with no known cure, dyslexia impacts one's ability to process and produce textual information [29, 28], and can lead to long-term social, emotional, and economic challenges such as less peer acceptance, poor self-image, lower educational attainment, and reduced employment opportunities [11, 27].

The rapid development and adoption of neural language technologies - such as the ChatGPT - makes them an important part of the information ecosystem and a promising assistive tool for people with dyslexia [35, 8]. However, most of existing neural language models have been developed and evaluated over typical text (e.g. WikiText [15], CommonCrawl[1]), with little consideration of dyslexia use case. The fairness and inclusivity of neural language technologies for users with dyslexia is thus largely underexplored.

In this paper, we present an evaluation of the current state-of-the-art machine translation (MT) models available via popular cloud services on dyslexia-style text. To evaluate potential biases

---

[1] https://commoncrawl.org/overview

Submitted to the 38th Conference on Neural Information Processing Systems (NeurIPS 2024) Track on Datasets and Benchmarks. Do not distribute.

presented in machine translation against dyslexia text, we perturbed the source text in WMT14 (en2fr) dataset [2] with synthetic dyslexia style writing errors, and benchmark the performance of four commercial machine translation systems using the perturbed data. Our results show all audited models - including advanced LLMs - struggle with dyslexia-style input text, making substantially more lexical and semantic mistakes. By varying the quantity and the types of dyslexia style errors injected into the original text, we also observe a near linear relationship between the amount of dyslexia errors and the decrease in performance for all services, especially for real-word errors such as the confusion of homophones [23, 24]. Our contribution to language technology, AI fairness, and accessibility research is two-fold: 1) Our findings uncover the disparities in the performance of commercial machine translation systems to translate dyslexia style text; 2) Our systematical approach in generating synthetic dyslexia datasets provides an useful instrument to further investigate the potential sources and mechanism for such disparities in typically "black-boxed" systems when real dyslexia datasets are scarce. As an early exploration in AI fairness and dyslexia, our work invites further investment and urgent attention from NLP researchers and commercial companies to develop inclusive and fair NLP models with people with dyslexia, a community deeply impacted by and highly experienced with language technologies.

## 2   Background and Related Work

### 2.1   Dyslexic Writing Style

There are many spellcheckers available that try to correct spelling errors but most are not specifically designed to address dyslexic-style writing. General use spell-checkers perform poorly when it comes to real-word errors [18] (e.g. form v.s. from). [25] found that this comprises of 17% of the errors made by English dyslexic people. The have been some efforts to create a dyslexia-style writing support tool from [21], [18] and [25]. Unfortunately, these systems are designed in an academic fashion and are not the most appropriate for a widespread writing style that is used in an everyday life. Previous work from [35] utilized data and writing from social media to give more relevance to everyday text. More recently, Goodman et al. [8] utilized a Large Language Model (LLM) to create an email-writing interface tool for users with dyslexia. In this study, we do not try to "correct" any dyslexic-style typographical errors, but to understand the capacity of commercial machine translation systems at handling text that contains this style of writing.

Dyslexia-style text has been categorized in previous works from [22]. The typographical errors presented were broken down into four categories. *Substitutions* were identified as letters that are changed with one another (reelly v. really). *Insertions* were counted where a letter is inserted (situartion v. situation) or where a word that was incorrectly split (sub marine v. submarine). *Deletions* was when a letter is omitted (approch v. approach). *Transpositions* were considered as two letters that were swapped and adjacent (artcile v. article). Using these categories [23] found that the substitutions were by far the most common type of dyslexic-style typographical error. It is to note that this was found on a Spanish corpus of hand written text by students with dyslexia.

Work from [19]created a large confusion set of words. This set consists of real word errors collected from dyslexic text [18] and also synthetically created samples that were used to test spell checkers. The set is a mix of homophones, substitutions, insertions, omissions (deletions) an transpositions. This is the most exhaustive set of dyslexic related errors that we were able to find. Previous work utilized synthetic dyslexia writing for neural translation models and demonstrated success in creating an assistive writing tool for dyslexic users on social media [35].

### 2.2   Subgroup Performance Disparities in AI Systems

Previous work from [4] and [5] has brought to light racial disparities in AI. Inequalities are often caused by lack of awareness in training data, fairness in training [31, 16] and other inclusive consid-erations. They found that people of the minority classes are the ones who suffer from shortcomings

of the machine learning models. Lack of data of different groups leads to the use of synthetic data like in [12] for stutter data.

Object-recognition systems displayed disparities in terms of income levels and geographies [7, 9]. Smaller subgroups are more at risk for poorer performances. Work from [10] identified key components (texture, occlusion and darker lighting) that lead to performance degradation of object-recognition systems in lower income levels/geographical areas and show that it is possible to mitigate these disparities.

Work from [30] spotlights the issue of the utilization of the English language when training models creating disparities. These disparities range from people being unable to utilize the models due to a language barrier to the models existing but not performing to par. Inequalities for resources, variation and performances is seen as the industry norm when we apply NLP to underrepresented communities.

Unfortunately, there has not been much work researching the affects of artificial intelligence on people with dyslexia. Researcher from [1] were able to gather an estimate of the amount of dyslexic text documents on the Web (0.005%). They deemed their estimate much lower than the corresponding number of dyslexic users (10-17%). If they considered spelling errors as dyslexic-style typographical errors, the number would increase to 0.2%. Therefor, it is likely that models trained on data from the Web is not reflective of the dyslexic population. This leads us to believe that the models are trained on "perfect" data that has been filtered through spell checkers. This potential bias is what is being studied in this paper for one NLP task.

### 2.3   NLP Model Evaluation and Benchmarking

Machine translation is a common NLP task where a source sentence is translated into a different target language. The *Machine Translation Foundation*[2] provides a new dataset yearly during the Conference on Machine Translation (WMT) to benchmark the performance of SOTA MT models on various translation tasks. Many language pairs with parallel data are provided in the WMT datasets, with public available source data and manually translated target references. We use WMT14 (en2fr) dataset for this study. It contains news articles in English as source data, together with parallel manual translation in French.

Following the breakthrough by Vaswani et al. [34], the transformer architecture has become increasingly popular for MT models. We assume that widely used translation services from major cloud service providers such as AWS, Google Cloud and Azure are utilizing this architecture. However, the exact model structure is not public information nor the data that is used in the training for these models. That means to understand and diagnose these systems, we have to rely on their APIs and translation outputs of a wide range of source sentences to shed light on the black box models.

## 3   Method

For our scope of work, we leveraged and modified the WMT14 (en2fr) [2] dataset to evaluate a machine translation task from English to French with injected synthetic dyslexic-style errors. We select machine translation for our exploratory evaluation because the task is well-defined, with well-established metrics and benchmarking datasets, as well as many popular consumer-facing applications and services such as Google Translate[3]. We also limit our initial benchmarking to the translation from English to French - two well-resourced languages for machine learning, to reduce potential confounding factors due to languages. In this section, we review how we created the synthetic dyslexic text corpora and the types of dyslexic writing errors injected. We then present and discuss the commercial machine translation services we evaluated using the synthetic dyslexic text. Finally, we describe the metrics and methods we utilized for benchmarking the performance of these services in both lexical and semantic dimensions.

---

[2]https://machinetranslate.org/about
[3]https://translate.google.com/

### 3.1 Simulating Dyslexia

The lack of large scale and publicly available dyslexic text corpus has been a bottleneck for dyslexia-related language technologies today [35, 8]. Direct collection of text written by people with dyslexia faces both ethical and practical challenges. As an "invisible" disability that is highly stigmatized, many people with dyslexia feel the pressure and need to conceal their dyslexia, spending extra efforts to proofreading their writing or avoiding to write at all [26]. Even if people with dyslexia consent to share their data, it is difficult to fully anonymize the data while preserving the unique and personal writing styles of dyslexia. Encouraged by the success of using synthetic disability data for data-intensive machine learning tasks [12, 35], we created a synthetic dataset of dyslexic text by injecting typical dyslexic writing errors into a popular MT benchmarking dataset, namely, the WMT14 (en2fr) test dataset [2]. Taking a similar approach proposed by Wu et al. [35], we perturbed the English source sentences with the following three synthetic errors that are frequent in dyslexic input text and less likely to be fixed by mainstream spellcheckers:

1. Letter confusion: substituting similar-looking or sounding letters (e.g. b v.s p). Letter confusion is reported as the most frequently occurred errors in dyslexic writing [23].

2. Homophone: replacing a word with its homophones. Phonetically similar sounding words are noted as another common but unique challenge for people with dyslexia [18], [23], and can potentially create issues for NLP models as this type of error is relatively rare in typical text used to train the models.

3. Confusion set: substituting a word with another word that are likely to be confused with by people with dyslexia (e.g. "*your*" and "*you*"). Previous work found confusion sets contribute a substantial percentage of dyslexic writing errors and are least likely to be caught by conventional spellcheckers [18, 24, 35].

To simulate letter confusion, we constructed a letter substitution dictionary in which each letter is associated with other letters people with dyslexia are often confused with [23]. The frequency of letter confusion is controlled by a parameter $p_l$, which represents the probability for letter confusion to occur in the original corpus. However, following empirical findings that letter confusion rarely occur at the beginning of a word [36, 20, 18], the substitution of the first letter would ignored 95% of the time during error injection. Also, to be consistent with the observations that multiple letter confusions are uncommon in dyslexic writing [23], we decreased the probability of another substitution happening by 90% for that same word after one substitution is made.

To simulate homophone errors, we constructed a homophone dictionary in which each word is associated with its phonetically similar sounding words. We leveraged free public resources such as the Homophone Finder website[4] to build the homophone dictionary. The frequency of homophone error is again controlled by a parameter $p_h$, which represents the probability for us to swap the current word with its homophone.

To simulate errors from confusion set, we constructed a dictionary using the confusion set identified by Pedler and Mitton [19]. This set contains around 6000 pairs of words that are likely to be confused with one another by people with dyslexia. The frequency of this type of error is controlled by $p_s$, representing the probability of a word being replaced by its paired word in the confusion set.

Examples of three types of injected errors are provided in Table 1. The original sentences are taken from WMT14 (en2fr). Note that the perturbed sentences with homophone and confusion set errors do not have misspellings but "real word errors" that are less likely to be detected and fixed by spellcheckers before being sent for machine translations [24].

With this in mind, we are able to modify the WMT14 (en2fr) test dataset with different $p$ values, resulting different quantities of dyslexic errors injected into original source data. In this paper, we focus on the percentage of words modified ranging from 10-20% as this follows findings from [23] in real world dyslexic text error rate.

---

[4] https://www.homophone.com

Table 1: Example synthetic dyslexic sentences with injected dyslexic writing errors

| Error Injection | Original Sentence | Perturbed Sentence |
|---|---|---|
| Letter Confusion | In Nevada, where about 50 volunteers' cars were equipped with the devices not long ago, drivers were uneasy about the government being able to monitor their every move. | In Nevada, where **abouf** 50 **wolunteers'** cars were equipped with **thi devoces** not **iong** ago, **driverc** were **nneasy** about the government being able to **mohitor** thein every **movo**. |
| Homophone | New York City is looking into one. | New York City is looking into **won**. |
| Confusion Set | "The gas tax is just not sustainable," said Lee Munnich, a transportation policy expert at the University of Minnesota. | "The gas tax is just **knot** sustainable," said Lee Munnich, **eye** transportation policy **export** at the University of Minnesota. |

## 3.2 Commercial Machine Translation Audit

We chose to evaluate SOTA models that are deployed across major cloud computing platforms namely, AWS, Azure and Google Cloud. Based on a survey from *Public First* 51% of business utilize cloud services, majority of which are customers of AWS, Azure and Google Cloud [5]. We also tested our dataset on GPT-3.5 (gpt-3.5-turbo-1106)[6] a large language model (LLM). For each one of these services, we tested the performance of document translation, and for GPT we did a sentence-level translation (document translation was not available). For document translation, we submitted text files to the services for translation. For sentence-by-sentence translation, we were able to call the OpenAI API with Python scripts. All of these platforms require payment for the use of the translation services. For Google Cloud, we used the Cloud Translation API, for AWS, we used the Amazon Translate service and for Azure, we used the Translator in the Cognitive Services. Once the text was received we were able to evaluate the text.

## 3.3 Evaluation Metrics

We evaluate the performance of commercial MT services over synthetic dyslexic text with both lexical and semantic metrics. While the lexical metrics - such as BLEU [17] and WER [32] - allow us to benchmark against position our results in relation to a wide range of MT models and tasks, the semantic metrics - such as BERT and LaBSE - help illustrate how dyslexia might impact the user experience of these MT services.

### 3.3.1 Lexical metrics

Lexical based metrics have been commonly used in the evaluation of machine translation systems [13]. One of the most popular lexical based metrics is Bilingual evaluation understudy (BLEU) [17], which is frequently used for in benchmarks and leaderboards. BLEU measures the n-gram similarity between MT output and the reference, and it is known for its simplicity, language-agnostics, and ability to measure both precision and fluency. BLEU score ranges from 0 to 1 where 1 indicates a perfect translation. State-of-the-art (SOTA) MT systems have reported BLEU score as high as 0.464 for WMT14 (en2fr) task [14], which could be considered as generally "high quality translations"[7]. In contrast, BLEU scores lower than 0.2 would be considered "hard to understand" and "almost useless".

The second lexical based metric we utilize is Word Error Rate (WER) [32], which measures the edit distance between MT output and the reference. As WER can be further broken down into the minimum number of word substitutions, insertions, and deletions required to convert the MT output

---

[5]https://awsus.publicfirst.co/
[6]https://platform.openai.com/docs/models/gpt-3-5-turbo
[7]BLEU Score Interpretations: `https://cloud.google.com/translate/automl/docs/evaluate`

to the reference sentence, this metric provides us additional insights into how the translation of perturbed dyslexic sentences differ from the original sentences. While WER can range from zero to infinity, a WER score higher than 0.5 generally suggests a poor performance.

### 3.3.2 Semantic Metrics

Since we are dealing with injected synthetic text, the lexical form of words are sometimes very similar (for example in third row of Table 1 we have "knot" v. "not"). The edit distance between the two samples is 1. However, the semantics of the words are completely different. This is where our lexical metrics would likely fail. In order to fairly compare the sentences, we introduce semantic calculations. The first method was using BERTScore [37] which computes a similarity score between 0 and 1 (where 1 is perfect) using contextual embeddings. The second evaluation metric we utilized was a language independent method LaBSE [6] where we were able to use the source English sentences from WMT directly for semantic comparison. We calculated the L2-norm of the sentence embeddings from LaBSE to get the similarity between the source English sentences (without injections) to the translations generated by the models. We called this the LaBSE score [8]. Same to the previous metric, the score ranges between 0 and 1 where 1 indicates identical sentences and meaning. We must note that a score of 1.0 requires the sentences to be syntactical identical. In other words, two sentences with identical meanings but different writing would not score 1.0, but very close to 1.0.

## 4 Results

### 4.1 Lexical Divergence

To measure how injected dyslexic errors influence translation results at a lexical level, we calculated the BLEU and WER scores using the French translation from perturbed English sentences as hypothesis and the original target sentences in French as references. We also calculated the BLEU and WER scores for the translations generated by each MT service over the original, unperturbed English data, as the baseline for our comparison.

We observed a SOTA level of performance in audited MT services at the baseline condition, with BLEU score ranging from 0.429 (GPT3.5) to 0.469 (Google). However, the performance consistently degrades as dyslexic style errors occur. Figure 1a shows a near linear drop in BLEU score, along with the increase of words perturbed with dyslexic errors. While GPT3.5 has the lowest baseline BLEU score, it is also least impacted by the increase of dyslexic errors. In contrast, the performance of Azure MT drops most drastically when encountering more dyslexic errors. In terms of error types, we notice that most services have more difficulties dealing with "real word errors" from homophone and confusion set, rather than syntactic errors like letter confusion, with Azure being the only exception. This observation is consistent with previous findings that real word errors in dyslexic writing pose greater challenges for NLP models [19, 24].

Similar trend is observed in WER scores. As shown in Figure 1b, for all audited services, their WER scores increase steadily as more synthetic dyslexic errors are injected into the source data. The slope of increase is greatest for homophone errors, and lowest for letter confusion. However, comparing to AWS and GPT3.5, Google and Azure seem to be particularly challenged by letter confusion errors, showing a degradation in translation quality almost as rapidly as when encountering synthetic real word errors. Further inspection of their translation results in this condition suggests that the MT services by Google and Azure are less likely to recover from a misspelled word, but tend to directly copy it in the translation. For example, when the baseline sentence "*The American **Civil** Liberties Union is deeply concerned*" is perturbed to become "*The American **Cavil** Liberties Union is deeply concerned*", Google and Azure would translate the perturbed sentence to "*L'American **Cavil** Liberties Union est profondément préoccupée*", with the misspelling "*Cavil*" preserved in the translation.

We also broke down the different types of edits used for calculating WER and inspect them separately. Figure 2 shows the breakdown of substitutions, insertions, and deletions in the translation of 20%

---

[8]https://huggingface.co/setu4993/LaBSE

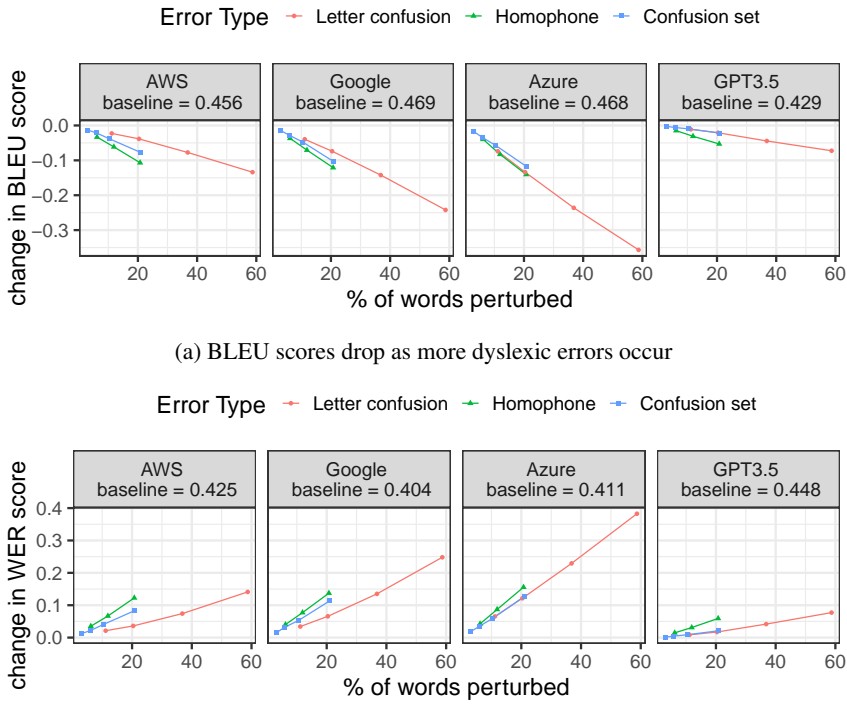

(a) BLEU scores drop as more dyslexic errors occur

(b) WER scores increase as more dyslexic errors occur

Figure 1: Change in lexical based metrics for all audited services. Baseline values indicate the metric score for unperturbed text, y-axis shows the change in corresponding metric compared to the baseline.

perturbed text from the reference. While the overall trends are similar for all MT services with three types of synthetic errors, we do observe some small difference in Azure and Google when handling letter confusion. These two services appear to make more deletions than insertions in their translation of text with letter confusion errors, suggesting potential loss of semantic information in the translation when source data contain significant amount of dyslexic misspellings. On the other hand, services like AWS and GPT3.5, despite more robust performance, tend to insert words in their translations. A deeper investigation on insertion errors found that articles are most often being inserted (see Figure 3 for the most commonly added words by AWS with 20% confusion set errors).

While GPT3.5 generally perform better with synthetic dyslexic text, its performance still declines and could sometimes make serious mistakes due to dyslexic errors. For example, when the baseline sentence "*The technology is there to **do it***" is perturbed to "*The technology is there to **do ti**. *", the translation by GPT3.5 diverges from "*La technologie est là pour **le faire***" to "*La technologie **le frappe de plein fouet***" ("technology hitting it head on").

## 4.2 Semantic Divergence

While lexical divergence, such as the insertion and deletion of particles, might not significantly impact the quality of translations, semantic change in the translation of dyslexic text from non-dyslexic text could have direct user experience consequences. While all audited services demonstrate high performance with unperturbed text at the semantic dimension (BERTScores and LaBSE scores all above 0.9), the semantic of the translation diverges as more dyslexic writing errors occur. As shown in Figure 4, both the BERTScore and LaBSE drops when the percentage of synthetic errors in text increases. Among all the audited services, the performance of Google and Azure declines most rapidly, while GPT3.5 maintains a relatively robust level of performance.

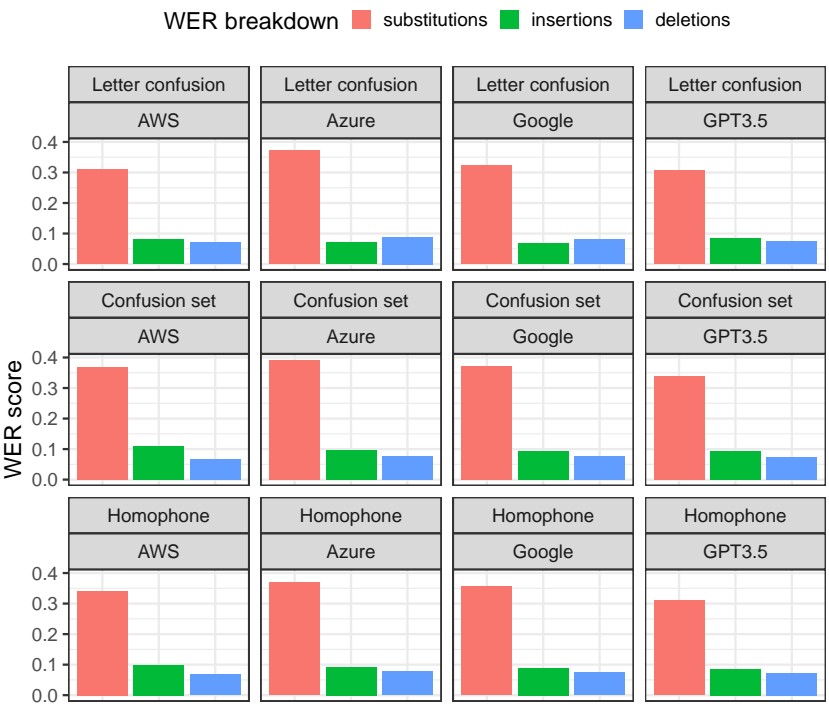

Figure 2: Breakdown of WER scores by edit type (20% word perturbed)

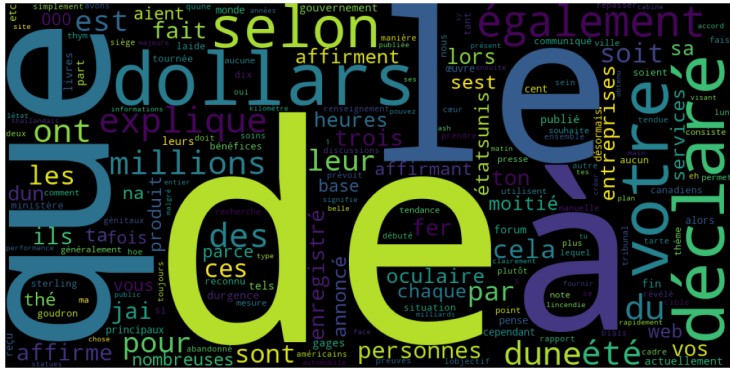

Figure 3: Word cloud of word confusion AWS (20% word modified)

Even if the sematic divergence is smaller comparing to the lexical divergence, the disparity between the baseline and text with 20% dyslexic errors is statistically significant, suggesting a clear gap in MT service quality for dyslexic users.

## 5   Discussion

Our results uncover potential disparities in the quality of MT services for people with and without dyslexia. As part of the cloud infrastructure, these services have been ubiquitously adopted as foundation for many other digital products and services. Our work shows how typical dyslexic writing errors could lead to the degradation of SOTA MT services. Even advanced LLMs, which have been believed as a solution for dyslexia, struggle with real word errors from homophones and

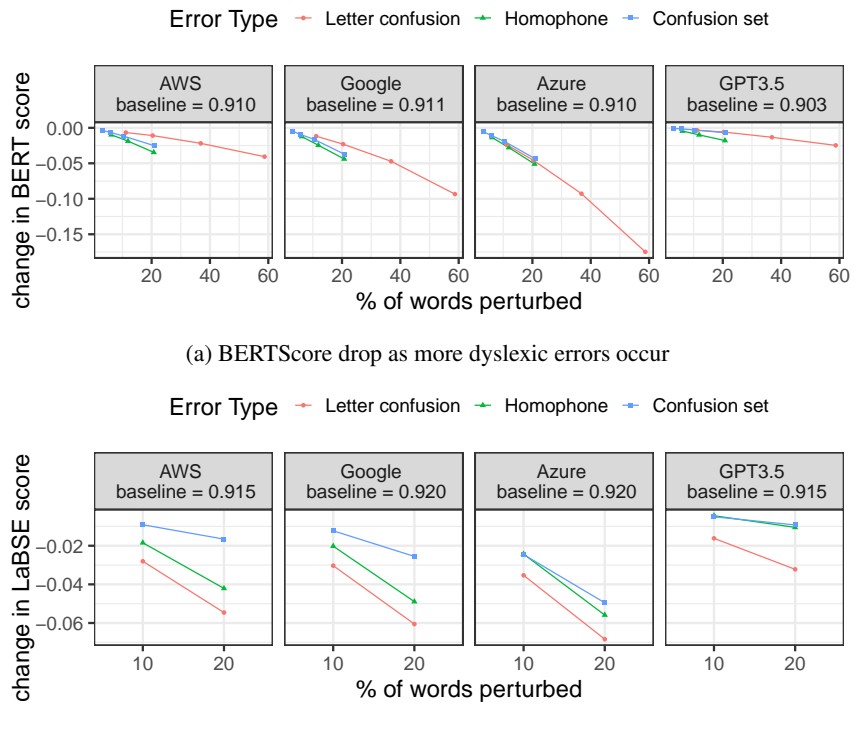

(a) BERTScore drop as more dyslexic errors occur

(b) LaBSE scores drop as more dyslexic errors occur

Figure 4: Change in semantic metrics for all audited services. Baseline values indicate the metric score for unperturbed text, y-axis shows the change in corresponding metric in comparison to the baseline.

confusion set. While LLMs are better than other services in terms of lexical and syntactic mistakes, they do still produce semantic divergence when translating dyslexic text, and such divergence could be even harder to be noticed by users with dyslexia, resulting in higher user risk and potentially worse experience in the long term.

# 6 Limitations and Future Work

Although we were able to experiment with a wide variety of configurations with the quantities and types of dyslexic writing errors, our synthetic datasets are nevertheless limited in their ability to capture the full heterogeneity of dyslexic writing. Like any other neurodivergence, dyslexia affects people differently: the way it manifests in writing differs across individuals and situations. More authentic, real world data from people with dyslexia is required to better represent this community in AI data in order to develop fair and inclusive NLP models for dyslexia. We also look forward to extend our methodology to other communities and application domains, making it easier to audit a wide range of AI models and services using synthetic data about marginalized, sensitive populations.

# 7 Conclusion

We proposed a novel method to generate synthetic dyslexia datasets and levaraged them to identify performance disparities in SOTA machine translation services for people with dyslexia. Our lexical and semantic metrics allow us to benchmark and better understand existing disparities. Our work highlights the importance of making NLP and AI more inclusive and equitable to communities most impacted by such technologies. We call for attention from language technology researchers and developers to close the equity gap for users with dyslexia.

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
