# Supplementary Materials

- URL to **code**: https://github.com/aimpowered/NLPdisparity
- URL to **dataset**: https://huggingface.co/datasets/gpric024/wmt14_injected_synthetic_dyslexia
- (can also be viewed on GitHub in the first link)
- URL to **croissant metadata**: https://huggingface.co/api/datasets/gpric024/wmt14_injected_synthetic_dyslexia/croissant
- **Author Statement:** We bear all responsibility in case of violation of rights and confirm that our data is shared under CC-BY-SA-4.0
- **Hosting and maintenance:** Our hosting and maintenance plan can be found in the following pages in the datasheet for datasets.

# Datasheet for datasets

## Motivation

**For what purpose was the dataset created? Was there a specific task in mind? Was there a specific gap that needed to be filled? Please provide a description.**

The dataset was modified to test the performance of state of the art machine translation models with dyslexia style text. Unfortunately, real text from people with dyslexia and the associated translations in other languages does not exist to our knowledge. Synthetic dyslexia style injections were made into the WMT14 Test set from Hugging Face (here).

**Who created the dataset (e.g., which team, research group) and on behalf of which entity (e.g., company, institution, organization)?**

The WMT14 dataset was created by Ondřej Bojar, Christian Buck, Christian Federmann, Barry Haddow, Philipp Koehn, Christof Monz, Matt Post, Lucia Specia in the *Findings of the 2014 Workshop on Statistical Machine Translation*. The modified version of the dataset was created by Gregory Price and Shaomei Wu from AImpower.org.

**Who funded the creation of the dataset? If there is an associated grant, please provide the name of the grantor and the grant name and number.**

The creation of the dataset was funded by the non-profit AImpower.org. The technical work was performed by Greg Price and Shaomei Wu as volunteers.

**Any other comments?**

No.

# Composition

**What do the instances that comprise the dataset represent (e.g., documents, photos, people, countries)? Are there multiple types of instances (e.g., movies, users, and ratings; people and interactions between them; nodes and edges)? Please provide a description.**

Each file within the dataset consists of a ".txt" or ".docx" file containing the translated sentences from AWS, Google, Azure and OpenAI. Each line represents a translated sentence. The file names indicate the type of synthetic error injection that was done to the English version and the associated injection probability. The "default" directory consists of the English versions that were submitted to the translation services. The "v1" and "v2" folder names can be ignored. File names and the folder name indicated the type of synthetic errors and the probability of injection. Each file is a version of modified WMT text with varying levels/types of injections. E.g. the file name "wmt14_en_p_homophone_0.2_p_letter_0.0_p_confusing_word_0.0" has a probability of 20% to inject a homophone error in a sentence, 0 % of replacing a letter with a confusing letter and 0% of replacing a word with another word from its confusion set. The dyslexic error injection process is explained in detail in our paper.

**How many instances are there in total (of each type, if appropriate)?**

There are 12 files associated with each translation service. Each file has varying levels of injection probability and injection type that can be identified using the file name.

**Does the dataset contain all possible instances or is it a sample (not necessarily random) of instances from a larger set? If the dataset is a sample, then what is the larger set? Is the sample representative of the larger set (e.g., geographic coverage)? If so, please describe how this representativeness was validated/verified. If it is not representative of the larger set, please describe why not (e.g., to cover a more diverse range of instances, because instances were withheld or unavailable).**

The original WMT-14 dataset was curated and released at the 9th Workshop on Statistical Machine Translation (2014). The validity of the original dataset was verified during its first release.The validity of our perturbed dataset is based on prior empirical work on dyslexia and dyslexic writing, which are referenced in our paper.

**What data does each instance consist of? "Raw" data (e.g., unprocessed text or images) or features? In either case, please provide a description.**

Each file consists of sentences. The English sentences were processed to ensure uniformity. The translated outputs from the services are unprocessed.

**Is there a label or target associated with each instance? If so, please provide a description.**

No, there are no labels. The associated information is in the file name as mentioned previously.

**Is any information missing from individual instances? If so, please provide a description, explaining why this information is missing (e.g., because it was unavailable). This does not include intentionally removed information, but might include, e.g., redacted text.**
No.

**Are relationships between individual instances made explicit (e.g., users' movie ratings, social network links)? If so, please describe how these relationships are made explicit.**
No, there are no relations identified other than where the text was collected from.

**Are there recommended data splits (e.g., training, development/validation, testing)? If so, please provide a description of these splits, explaining the rationale behind them.**
No, no training has been performed on this data.

**Are there any errors, sources of noise, or redundancies in the dataset? If so, please provide a description.**
Yes, there are 4 instances where GPT outputted the following response: "I'm sorry, but I cannot provide a translation for the sentence you provided as it appears to be a combination of misspelled words and does not make sense. If you could please provide a correctly spelled and coherent sentence, I would be happy to assist you with the translation". We nevertheless considered these sentences as the translated output as they are what a user with dyslexia would receive when submitting a translation request to GPT.

**Is the dataset self-contained, or does it link to or otherwise rely on external resources (e.g., websites, tweets, other datasets)? If it links to or relies on external resources, a) are there guarantees that they will exist, and remain constant, over time; b) are there official archival versions of the complete dataset (i.e., including the external resources as they existed at the time the dataset was created); c) are there any restrictions (e.g., licenses, fees) associated with any of the external resources that might apply to a dataset consumer? Please provide descriptions of all external resources and any restrictions associated with them, as well as links or other access points, as appropriate.**
The dataset is self-contained, however, if someone ran the same experiment on the translation services it is possible that the outputs are different due to updates to the audited translation services.

**Does the dataset contain data that might be considered confidential (e.g., data that is protected by legal privilege or by doctor– patient confidentiality, data that includes the content of individuals' non-public communications)? If so, please provide a description.**
No.

**Does the dataset contain data that, if viewed directly, might be offensive, insulting, threatening, or might otherwise cause anxiety? If so, please describe why.**
No, not that we are aware of, either was mentioned in the original curation of the WMT-13 dataset.

# Collection Process

**How was the data associated with each instance acquired? Was the data directly observable (e.g., raw text, movie ratings), reported by subjects (e.g., survey responses), or indirectly inferred/derived from other data (e.g., part-of-speech tags, model-based guesses for age or language)? If the data was reported by subjects or indirectly inferred/derived from other data, was the data validated/verified? If so, please describe how.**

The collection of the original parallel data (EN-FR) was done previously by Ondřej Bojar, Christian Buck, Christian Federmann, Barry Haddow, Philipp Koehn, Christof Monz, Matt Post, Lucia Specia in the *Findings of the 2014 Workshop on Statistical Machine Translation*. 1500 sentences were collected from English news sources. Another 1500 were collected from news sources in a different language and translated by humans to English which created the 3000 sentence test set. The exact sources can be found in the linked paper.

**What mechanisms or procedures were used to collect the data (e.g., hardware apparatuses or sensors, manual human curation, software programs, software APIs)? How were these mechanisms or procedures validated?**

Please refer to *Findings of the 2014 Workshop on Statistical Machine Translation* for the collection of the non injected data. For the translated outputs, the Amazon Translate service on AWS was used, Google Document translate was used, for Azure, the Azure AI services Translator was used and for GPT, gpt-3.5-turbo-1106 available via the OpenAI API was used.

**If the dataset is a sample from a larger set, what was the sampling strategy (e.g., deterministic, probabilistic with specific sampling probabilities)?**

The test set of WMT14 was used, given our computational resources.

**Who was involved in the data collection process (e.g., students, crowdworkers, contractors) and how were they compensated (e.g., how much were crowdworkers paid)? Over what timeframe was the data collected? Does this timeframe match the creation timeframe of the data associated with the instances (e.g., recent crawl of old news articles)? If not, please describe the timeframe in which the data associated with the instances was created.**

The data collection was done by the authors of *Findings of the 2014 Workshop on Statistical Machine Translation*. The collection and creation of the perturbed data was done by Greg Price and Shaomei Wu from AImpower.org.

**Were any ethical review processes conducted (e.g., by an institutional review board)? If so, please provide a description of these review processes, including the outcomes, as well as a link or other access point to any supporting documentation.**

No.

# Preprocessing/cleaning/labeling

**Was any preprocessing/cleaning/labeling of the data done (e.g., discretization or bucketing, tokenization, part-of-speech tagging, SIFT feature extraction, removal of instances, processing of missing values)? If so, please provide a description. If not, you may skip the remaining questions in this section.**
The only text preprocessing revolved around ensuring punctuation was identical to the original data after injecting dyslexia style writing. Also, failed outputs from the OpenAI were kept as previously mentioned.

**Was the "raw" data saved in addition to the preprocessed/cleaned/labeled data (e.g., to support unanticipated future uses)? If so, please provide a link or other access point to the "raw" data.**
Original raw data can be found on Hugging Face.

**Is the software that was used to preprocess/clean/label the data available? If so, please provide a link or other access point.**
Yes. The code to generate the synthetic dataset can be found at our public github repo.

**Any other comments?**
No.

# Uses

**Has the dataset been used for any tasks already? If so, please provide a description.**
Our dataset with injected synthetic dyslexic errors is the first of its kind and has not been shared anywhere else.

**Is there a repository that links to any or all papers or systems that use the dataset? If so, please provide a link or other access point.**
The WMT14 test set with injections has not been used anywhere else as we modified it ourselves.

**What (other) tasks could the dataset be used for?**
The dataset should be used to measure machine translation models on dyslexic style text. Further work needs to be done to see if the set is comprehensive enough to be used for training.

**Is there anything about the composition of the dataset or the way it was collected and preprocessed/cleaned/labeled that might impact future uses? For example, is there anything that a dataset consumer might need to know to avoid uses that could result in unfair treatment of individuals or groups (e.g., stereotyping, quality of service issues) or other risks or harms (e.g., legal risks, financial harms)? If so, please provide a**

**description. Is there anything a dataset consumer could do to mitigate these risks or harms?**

The dataset consists of **synthetic** dyslexic text with dyslexia-style writing errors injected based on previous research on dyslexia writing styles. The detailed methodology that can be found in our paper. However, the data should not be interpreted as real dyslexic text.

**Are there tasks for which the dataset should not be used? If so, please provide a description.**

Any future work needs to consider that the dataset consists of synthetic injections and should not be presented as real dyslexic text. Other than that, there are no restrictions on how the data should or should not be used.

**Any other comments?**

No.

# Distribution

**Will the dataset be distributed to third parties outside of the entity (e.g., company, institution, organization) on behalf of which the dataset was created? If so, please provide a description.**

Yes the dataset will be publicly available.

**How will the dataset be distributed (e.g., tarball on website, API, GitHub)? Does the dataset have a digital object identifier (DOI)?**

The dataset is available via Hugging Face: [https://huggingface.co/datasets/gpric024/wmt14_injected_synthetic_dyslexia](https://huggingface.co/datasets/gpric024/wmt14_injected_synthetic_dyslexia) . Its DOI is 10.57967/hf/2476.

**When will the dataset be distributed?**

The dataset is now available as of 2024.

**Will the dataset be distributed under a copyright or other intellectual property (IP) license, and/or under applicable terms of use (ToU)? If so, please describe this license and/or ToU, and provide a link or other access point to, or otherwise reproduce, any relevant licensing terms or ToU, as well as any fees associated with these restrictions.**
**Have any third parties imposed IP-based or other restrictions on the data associated with the instances? If so, please describe these restrictions, and provide a link or other access point to, or otherwise reproduce, any relevant licensing terms, as well as any fees associated with these restrictions.**

The license is cc-by-sa-4.0 (ATTRIBUTION-SHAREALIKE 4.0 INTERNATIONAL).

**Do any export controls or other regulatory restrictions apply to the dataset or to individual instances? If so, please describe these restrictions, and provide a link or other access point to, or otherwise reproduce, any supporting documentation.**

No.

**Any other comments?**
No.

# Maintenance

**Who will be supporting/hosting/maintaining the dataset?**
AImpower.org will be supporting the dataset via GitHub and Hugging Face.

**How can the owner/curator/manager of the dataset be contacted (e.g., email address)?**
The creators Greg Price (gpric024@uottawa.ca) and Shaomei Wu (shaomei@aimpower.org) can be contacted through emails.

**Is there an erratum? If so, please provide a link or other access point.**
No.

**Will the dataset be updated (e.g., to correct labeling errors, add new instances, delete instances)? If so, please describe how often, by whom, and how updates will be communicated to dataset consumers (e.g., mailing list, GitHub)?**
No, there are no plans to update or modify the dataset.

**If the dataset relates to people, are there applicable limits on the retention of the data associated with the instances (e.g., were the individuals in question told that their data would be retained for a fixed period of time and then deleted)? If so, please describe these limits and explain how they will be enforced.**
N/A

**Will older versions of the dataset continue to be supported/hosted/maintained? If so, please describe how. If not, please describe how its obsolescence will be communicated to dataset consumers.**
There is only one version of the dataset and it is static. No changes will be made.

**If others want to extend/augment/build on/contribute to the dataset, is there a mechanism for them to do so? If so, please provide a description. Will these contributions be validated/verified? If so, please describe how. If not, why not? Is there a process for communicating/distributing these contributions to dataset consumers? If so, please provide a description.**
Others may contribute and are encouraged to, they should contact the original authors about incorporating any changes.

**Any other comments?**
No.