# OpenReview forum: "Lost in Translation: Benchmarking Commercial Machine Translation Models for Dyslexic-Style Text"
_NeurIPS.cc/2024/Datasets_and_Benchmarks_Track — Submitted to NeurIPS 2024 Track Datasets and Benchmarks_

### Official Review · Reviewer_UQGJ · 2024-07-25
**Research on dyslexia is inspiring but using synthetic data for evaluation is not convincing**

**Rating:** 2
**Confidence:** 5
**Correctness:** I'm not convinced by the adoption of …
**Clarity:** yes, the writting in general is good.

**Review:**

The paper is generally well written with good clarity. The used data construction method is not novel and the developed evaluation set is based on existing WMT14 parallel data.

**Strengths:**

Showed that existing models may not be inclusive enough for people with dyslexia.

**Additional Feedback:**

see Opportunities For Improvement.

**Documentation:**

Yeah, the authors provided a supplementary about the dataset documentation.

**Limitations:**

The authors discussed the limitation.

**Opportunities For Improvement:**

* While using synthetic data for training like [35] is popular, using such data for evaluation is not convincing as they don't reflect the real world and are very likely injected with biases. It’s more convincing to construct an evaluation benchmark like [35] on top of real data.
* WMT14 En-Fr is not constructed by translating English to French; parts of it are translated from French to English.
*  Machine translation often adopts BLEURT for semantic evaluation, rather than BERTScore.
* It would be great to test on multiple language pairs.
* In conclusion section, the authors claimed “We proposed a novel method to generate synthetic dyslexia datasets”, but this method has already been proposed in [35].

**Relation To Prior Work:**

The authors discussed the background and related work

**Summary And Contributions:**

This paper examines how dyslexic-style text (which often contains dyslexic errors) affects the current machine learning models. The authors focus on machine translation and developed a benchmark based WMT14 En-Fr using  synthetic dyslexic data construction. They find that the popular commercial translation systems are not robust enough to dyslexic errors, which often degrade significantly.

---

> ### Author Rebuttal · Authors · 2024-08-19
>
> Thank you very much for your review. We appreciate your recognition of the significance of the topic and would like to provide additional contexts and thoughts on our methodological selection and execution.
>
> 1. Use of synthetic data for bias detection and diagnosis.
>
> Given the lack of publicly available real dyslexia dataset and the privacy and operational challenges to collect them, our method is intended to make a first step in this topic in an attempt to highlight (and eventually narrow) the user experience gap for dyslexic and non-dyslexic users.
>
> We provided more details and discussions on dyslexic data scarcity and challenges to collect them in our response to Reviewer BrsV02 above (see “Methodological novelty” section). We also want to highlight some unique benefits of synthetic data in bias detection and diagnosis. In particular, the ability to construct a wide range of conditions through programmatically controlling and tuning the types and rates of different dyslexic writing errors provide a good instrument for researchers to probe and learn the inner workings of the audited commercial - and often blackboxed - systems based on how these systems respond to different conditions. We hope our work demonstrates the potential of this method and encourages the adoption of it in similar work on NLP system fairness.
>
> 2. Potential extension of this work
>
> Thank you for clarifying the bi-directional construction of the WMT14 En-Fr dataset. Our methodology can be extended to injecting dyslexic errors into French as well, with the availability of French letter confusion sets, homophones, and word confusion sets. We also acknowledge that dyslexia style errors can different from language to language, and the construction of language specific letter/word confusion sets would require a non-trivial amount of work and domain knowledge.
>
> We also appreciate the suggestion of BLEURT metrics and will add it to our revision.
>
> 3. Tune down methodological contribution claim
>
> We understand your concern around our claim on methodological novelty in the conclusion section and agree that the claim is too general. As mentioned in our response to Reviewer ix6v05, we will tune down the claim and reword it as “we proposed a method to simulate and extrapolate dyslexia style writing - a type of data that is currently underrepresented in the training and evaluation of NLP systems”.

---

### Official Review · Reviewer_BrsV · 2024-08-02

**Rating:** 4
**Confidence:** 4
**Clarity:** Yes, the paper is clearly and compreh…

**Review:**

This paper introduces a straightforward method for automatically constructing a benchmark to evaluate the translation performance of existing systems on dyslexia data. The evaluation results on four state-of-the-art systems highlight significant challenges and provide further analysis of various error types.

Quality: The paper lacks rigorous verification for the correctness of the benchmark construction method. The experimental results are not supported by human evaluation, and there is insufficient analysis of the underlying causes of errors. Overall, the findings are not convincing.

Clarity: The writing is clear and accessible, effectively conveying the authors' completed work. However, the paper lacks required documentation elements for this track, which should be addressed.

Originality: The proposed method for creating dyslexic data relies on previous approaches and does not adequately highlight the innovations compared to existing methods. While the paper confirms that previous findings are applicable to machine translation systems, it does not provide many new insights specific to these systems.

Significance: The research topic is valuable, but the scope is limited to a narrow range of downstream applications. Furthermore, the discussion is not thorough enough for the areas covered. Additional content and analysis are needed to enhance the paper's impact.

**Strengths:**

1. Contribution to Fairness and Inclusivity in AI Systems: It addresses the critical yet underexplored issue of AI system unfairness arising from distributional differences between training data and real-world needs, with a particular focus on dyslexic users. It introduces an automated benchmark construction method that facilitates large-scale evaluation. The study demonstrates the suboptimal performance of multiple state-of-the-art MT systems in handling dyslexic-style text.

2. Focus on Dyslexic Minority Groups: It advocates for the development of AI systems that are more suitable for dyslexic individuals, highlighting significant ethical and social implications.

3. Clarity of Writing: The writing is clear and easy to understand, clearly presenting the proposed methods and conclusions.

**Additional Feedback:**

1. The availability of real dyslexic datasets is insufficient. However, if solely for evaluation, would using real data be feasible? This could serve as a ground truth for comparison, demonstrating the effectiveness of the constructed benchmark.

2. Compared to some open-source commercial MT systems, I would be more interested in how different LLMs with varying capabilities handle this type of noise. Do stronger LLMs exhibit greater robustness?

3. Can simple prompts mitigate this issue? For example, informing LLMs of the noise presence in source text beforehand or having LLMs correct the noise before translation. Additionally, to what extent can few-shot learning improve the system? I would expect to see insights on improving the system in the paper.

4. For machine translation systems, human evaluation is essential. Could you use a human evaluation framework like MQM, even on a small subset? Showing specific error types and proportions could provide valuable guidance for subsequent system improvements.

**Correctness:**

The paper does not provide sufficient evidence to support the correctness of the proposed benchmark. The final experimental results are validated across multiple metrics, but for MT systems, human evaluation is expected.

**Documentation:**

No, the paper does not include any appendices to address these issues.

**Opportunities For Improvement:**

1. Limited Novelty: The attention to dyslexic users is not a novel concept. The study's conclusions, particularly the struggle with dyslexic errors, especially real-word errors, echo previous findings. The primary distinction lies in its application to MT systems, but it does not present many new or meaningful insights.

2. Quality Verification of Generated Dyslexic Data: It is crucial to verify the quality of the constructed data and its ability to reveal the challenges faced by dyslexic users. Specifically, it is necessary to verify whether the generated sentences are excessively noisy and incomprehensible to humans, or if, as mentioned in related work, they provide more relevance to everyday text. A comparison with existing dyslexic data collection methods is also necessary.

3. Additional Metrics Needed: Although the paper employs four metrics, incorporating results on COMET, a recognized benchmark for semantic divergence, would enhance its credibility. Moreover, human evaluation is highly valuable for this study. It can validate the conclusions drawn and provide further analysis on why translation systems perform poorly.

4. Limit Coverage: The paper has limited content, presenting a relatively simple data construction method and reiterating the impact of dyslexic errors on performance. More extensive content is expected, such as analyzing more LLMs across various dimensions or providing insights for further improvements.

**Relation To Prior Work:**

The paper thoroughly discusses related work and its contributions. However, a concern is that although the paper extensively cites related papers in dyslexic data construction, it does not clearly explain the differences and advantages of its construction method compared to others, lacking proof of its innovation.

**Summary And Contributions:**

Summary:

This submission presents a construction method for synthetic dyslexia data to investigate the performance of commercial machine translation (MT) services in translating dyslexic-style text. The study evaluates Google Translate, AWS Translate, Azure Translator, and GPT-3.5. Dyslexia introduces unique writing errors, such as letter confusion, homophone mistakes, and word confusion sets, which are not typically addressed in existing MT models, thereby affecting the fairness of AI systems. To quantitatively demonstrate this, the study injects synthetic dyslexic errors into the WMT14 en2fr dataset and benchmarks the performance of these MT services.

Contributions:

1. Synthetic Dataset Creation: It develops a method to inject typical dyslexic writing errors into standard datasets.

2. Comprehensive Evaluation: It benchmarks major commercial MT services on dyslexic-style text, highlighting their struggle with dyslexic errors, especially real-word errors.

3. Call to Action: It emphasizes the need for more inclusive NLP models to better serve dyslexic users, ensuring AI system fairness and inclusivity.

---

> ### Author Rebuttal · Authors · 2024-08-19
>
> Thank you reviewer BrsV02 for your insightful review, we particularly appreciate your summarization of our methodological contribution and your suggestions on potential solutions for addressing dyslexia biases in LLMs. We will definitely explore these solutions in the next phase of this work. We also want to take this opportunity to share our thoughts and additional details regarding your concerns on methodological novelty, evaluation metrics, and the coverage of this work.
> 1. Methodological novelty
>
> We agree with you that real text written by people with dyslexia would be ideal, and arguably better suited for evaluation purposes. However, the collection and publication of such dataset has been challenging, for reasons including: 1) as a hidden disability that has only been understood more recently, dyslexia has been under-diagnosed for most adults with this condition; 2) the heterogeneity within dyslexia results in a wide range of patterns and challenges both reading and writing; 3) people with dyslexia often feel pressure to mask dyslexia in their public writing due to social expectations and stigma. As a result, it is difficult to confidently classify whether a piece of text is written by someone with dyslexia without knowing the identity of the author. Collecting and sharing private, unpolished writings from people formally diagnosed with dyslexia would trigger serious privacy concern as it risks revealing sensitive, stigmatized identities.
>
> Therefore, existing dyslexia writing data discussed in the literature are mostly collected manually, from students with dyslexia within the school setting (e.g. homework assignments), providing a rather limited representation of dyslexia writings in the real world. Even such data was hardly published. To the best of our knowledge, there are no publicly available real dyslexic datasets in English today (the dataset used by Pedler and Mitton to extract the confusion sets is no longer available at the link they provided in their paper).
>
>
> While we echo strongly with the reviewer on the need and value of authentic dyslexic writing data from and by people with dyslexia, we do not think the lack of real dyslexic datasets - and the challenge to collect them - should block the research community from investigating and making progress on this topic. The synthetic dyslexic data, as generated by our method, is not meant to replace the need for real dyslexic data but to open up the space for further investigation. As our synthetic data captured three specific yet common dyslexic writing patterns, our results illustrate the gap in MT service performance in these limited and simplified conditions, pointing to potentially greater challenges for real dyslexic writings.
>
> While our method for generating synthetic dyslexic data is built upon existing work such as [35], the novelty comes with new and important applications of synthetic data generation: beyond generating training data, we demonstrate the value and feasibility of this method in detecting and diagnosing system biases for under-represented groups - an increasingly urgent and important issue for the AI community.
>
> 2. Evaluation Metrics
>
> We appreciate the suggestion for COMET and will add it for measuring the semantic divergence in our revision.
>
> We also agree with the reviewer that human evaluation - especially by dyslexic users - would be extremely valuable and will design and implement it for the next phase of this work.
>
> 3. Coverage of this work
>
> We agree with the reviewer that the inclusion of more and stronger LLMs across more dimensions will yield interesting and deeper insights on this topic. However, we also want to clarify that our current prioritization of popular, commercial MT services and products was influenced by the amount of user impact these services and products have on people with dyslexia today. Our work is intended to uncover systematic, large-scale discriminations and potential harms towards people with dyslexia by these systems today, in the hope of raising attention and awareness of this issue within and beyond the NLP community.
>
> Lastly, with regards to your comment: "the paper lacks required documentation elements for this track, which should be addressed”, could you elaborate further please? We have provided a supplementary materials pdf which contains links to code, datasets and other related information, and would be happy to provide any additional documentations and details missed in current submission.

---

### Official Review · Reviewer_ix6v · 2024-08-05
**Review of Lost in Translation: Benchmarking Commercial Machine Translation Models for Dyslexic-Style Text**

**Rating:** 4
**Confidence:** 4
**Correctness:** see above
**Clarity:** see above

**Review:**

The paper addresses an important issue but has several shortcomings, primarily in the evaluation of only commercial MT models and the introduction of synthetic noise to the dataset. I am also a bit skeptical about the statement, "We proposed a novel method to generate synthetic dyslexia datasets," as the authors take an existing dataset and merely add noise based on established methods in the literature.

**Strengths:**

- Interesting and important topic.
- Usage of established MT dataset.

**Additional Feedback:**

-

**Documentation:**

see above

**Ethics:**

-

**Limitations:**

see above

**Opportunities For Improvement:**

The third type of error, "Confusion set," would benefit from more explanation.

The current evaluation of MT primarily relies on neural metrics like COMET, which are missing from the evaluation.

The evaluation is missing open-source models. While there is some value in evaluating commercial models, it feels incomplete without including open-source alternatives.

Figure 1: It would be helpful to see absolute scores in addition to the metric drop.

Line 227: The reporting of BLEU scores is unusual; they are often written from 0 to 100.

How should we interpret Figure 3?

**Relation To Prior Work:**

see above

**Summary And Contributions:**

The paper aims to evaluate how well commercial MT systems handle text with errors commonly made by individuals with dyslexia. The research is relevant, focusing on inclusivity in NLP. The paper's contributions include the generation of synthetic dyslexic text using three perturbation techniques and the evaluation of commercial MT models.

---

> ### Author Rebuttal · Authors · 2024-08-19
>
> Thank you  for the valuable feedback and for acknowledging the importance of the topic we explored in this work. We appreciate your comments and suggestions on research contribution, clarity, and scope, and would like to address them below.
> 1. Clarify our  contribution
> We understand your concern regarding the statement “We proposed a novel method to generate synthetic dyslexia datasets” and would be happy to tone down the claim to “we developed  a systematic method to inject typical  dyslexic writing errors into standard NLP datasets,  making it possible to increase the representation of dyslexic text in NLP systems in an efficient, privacy-preserving way.” This statement is adapted from the comment by Reviewer BrsV02 and we found it accurately and adequately describes our methodological contributions.
> 2. Clarify methods and results
>       1. The third type of error known as “confusion set” is based on the  previous work of Jennifer Pedler and Roger Mitton (http://www.lrec-conf.org/proceedings/lrec2010/pdf/122_Paper.pdf). They used real-word errors from the writings of dyslexic students and came up with a list of sets of words that are likely to be confused by people with dyslexia. Examples include [ace, ache, acre, axe] and [chat, chart, cheat, chit].It is worth noting that while some of the words in the same confusion set sound similar, they are not strictly homophones and can be confused due to similar spellings.We thus chose this type of errors as a complement to letter and homophone confusions.
>       2. For Figure 1, we will add additional graphs to show absolute scores.
>       3. For Line 227, thank you for your suggestion. We will report BLEU scores on the 0-99 scale in our revisions.
>       4. We will elaborate in our revision that Figure 3 demonstrates the frequency of words that are most commonly inserted, thus providing some intuition on how the MT systems are underperforming with dyslexic input. For example, we noticed that articles (“déterminants” in French) are added to create structurally correct sentences but  result in a deviation of the original meaning of the sentences.
> 3. Extend the scope of our work
>       1. Additional metrics: thank you for your suggestion of COMET, we will definitely add it to the next phase of this work.
>       2. Additional models: Thank you for your suggestion to include opensource MT models, we agree that it would further strengthen this work and gain even deeper insights into the inner works of MT systems in relation to dyslexia. However, we do want to clarify that this work is positioned as an exploration towards uncovering - and hopefully addressing - a complex technical and social challenges for dyslexia users of NLP systems, and we prioritize MT *services* and *products* such as Google Translate and ChatGPT as they have been ubiquitously deployed and used by millions of people everyday, including people with dyslexia. Considering their market dominance and product growth, these MT services have significant, yet unsatisfying impact on the lives of users with dyslexia. We hope our work sheds light on the potential user experience challenges for dyslexic users with current MT products and services, a topic that, to the best of our knowledge, has been studied by the NeurIPS community before.

---

### Decision · Program_Chairs · 2024-09-26

**Decision:**

Reject

**Comment:**

The paper explores how commercial machine translation (MT) systems handle text containing errors commonly made by individuals with dyslexia, emphasizing the importance of inclusivity in natural language processing (NLP). The study's contribution lies in generating synthetic dyslexic text through three perturbation techniques and evaluating the performance of popular commercial MT models, such as Google Translate, AWS Translate, Azure Translator, and GPT-3.5, on this data.

Reviewers acknowledged the significance of this study in promoting inclusivity in AI systems. Dyslexia introduces specific writing errors, like letter confusion and homophone mistakes, which current MT models often overlook. These limitations impact the fairness of AI, highlighting the need for more inclusive models. By addressing this issue, the paper contributes to making AI technologies more accessible and equitable for individuals with dyslexia.

However, several critical issues were raised. The study’s novelty is limited, as the method lacks significant innovation compared to existing approaches, particularly in comparison to reference [35]. Furthermore, although the study aims to explore how MT systems handle errors made by individuals with dyslexia, it relies on synthetic data, which, while useful, does not fully reflect the real-world errors made by dyslexic individuals. Additionally, the quality of the generated data has not been verified, raising concerns about its accuracy and relevance.

In the rebuttal, the authors argue that their novelty lies in the important application of synthetic data generation, extending beyond generating training data. They acknowledge that real text from dyslexic individuals would be ideal, but collecting such data is challenging. They assert that the absence of real dyslexic datasets should not prevent the research community from making progress in this area.

Overall, I agree with the reviewers that while the study addresses an important issue, the original contribution may not be sufficient enough for publication. The reliance on synthetic data and limited novelty in the approach reduces the impact of the findings.